# The Digital Automatic Coupler (DAC): An Effective Way to Sustainably Increase the Efficiency of Freight Transport in Europe

**Luciano Cantone** [1,*], **Thierry Durand** [2], **Andrea Ottati** [3], **Giulia Russo** [4] **and Roberto Tione** [5]

1 Department of Enterprise Engineering, University of Rome Tor Vergata, 00133 Rome, Italy
2 SNCF Voyageurs, Centre D'ingénierie du Matériel (CIM), 72100 Le Mans, France
3 FSI/Trenitalia, Technical Direction, 50129 Florence, Italy
4 UIC, Rail System Department, 75015 Paris, France
5 Wabtec, Faiveley Transport, 10045 Turin, Italy
\* Correspondence: luciano.cantone@uniroma2.it; Tel.: +39-06-7259-7133

**Abstract:** This paper employs and improves *TrainDy* software to simulate the in-train forces of single and coupled freight trains with a Digital Automatic Coupler (DAC), having a length from 740 m to 1500 m and hauled mass between 1600 t and 8000 t. Software improvements have focused on the ability to natively simulate the electro-pneumatic (EP) brake, which will be a feature of the future DAC. The various types of DAC and their main features are outlined in the paper. The study shows that it is also possible to increase the trains hauled mass by using radio communication between traction units (TUs). The "relative approach" method is used to demonstrate the safety of these new trains. The paper's main results are reported in the conclusions and demonstrate the effectiveness of both technologies in increasing freight transport efficiency, even when applied to coupled trains where one train is "classic" and the other is equipped with mechanical DAC.

**Keywords:** *TrainDy*; DAC; freight; longitudinal train dynamics (LTD)

## 1. Introduction

It is in the interest of Railway Undertakings (RUs) to increase freight train efficiency safely. Considering only the longitudinal train dynamics (LTD), i.e., the motion of the vehicles along the track (including the relative motions), there are several ways to accomplish this task. By increasing the train speed, goods are delivered in less time; since the train braking distance increases with its speed, this method is effective if the "braked weight" increases. Towards this end, wagons with improved braking features are needed, or existing wagons should be revamped; alternatively, the train braking distance should be increased. Another way is increasing the wagon mass: more goods are carried within the same time. Currently, in many European countries, the maximum mass per axle is 22.5 T; increasing this mass requires updates similar to those for the train speed increase and an updating of railway infrastructure to carry the higher wagon mass. A third way is the increase of train hauled mass by increasing the train length: the efficiency increases in a way similar to the previous method. The train length increment requires a railway infrastructure update (e.g., the "sidings" renewal to park the trains temporarily) and a technological improvement of the rolling stock to avoid train derailments or disruptions caused by excessive in-train forces. Among these methods, this paper deals with increasing train length and wagon innovations aiming to increase the permissible longitudinal compressive forces and reduce the in-train forces.

Most freight wagons are equipped with a simple air brake [1]: this system is characterized by asynchronous braking among the wagons, and those closer to the Traction Units (TUs) brake first. This asynchronous braking causes in-train forces during brake

applications [2] (see also [3] for a review on the topic). If the in-train compressive forces or the Longitudinal Compressive Forces (LCF) applied to a general wagon exceed its permissible value [4,5], the wagon would derail. To avoid this event, which is dangerous both for people and for goods and infrastructure, and to increase the efficiency of freight transport, a solution recently investigated in Europe by the Shift2Rail Joint Undertaking initiative is radio communication among the TUs through two "open call" projects, DYNAFREIGHT and Marathon2Operation (M2O) [6,7].

Recently, a new, ambitious program has been launched in Europe to improve the railway sector: Europe's Rail [8,9]. Among the different activities, this program envisages the renewal of freight wagons by implementing the Digital Automatic Coupler (DAC). Depending on its implementation level (briefly discussed in Section 2), a DAC can increase the wagon's permissible LCF (PLCF) and alter the in-train forces by substituting the side buffers and the draw gear with an automatic coupler (see, for example, [10–12] about the effects of central couplers). Moreover, a DAC can significantly reduce the in-train forces by introducing the electro-pneumatic brake (EP) or the electronically controlled pneumatic (ECP) brake (see [13,14]).

Since freight trains have several heterogeneities in terms of length, hauled mass, wagons type, and payload, the UIC IRS 40421 [15] provides a general method based on the "relative approach" for assessing the safety of a train against the risk of derailment; the method is also referenced by the [16]. This paper applies this method to evaluate the benefits of two technologies capable of effectively increasing freight efficiency: the radio communication between traction units and the DAC. This evaluation is based on *TrainDy* numerical simulations. In 2009, UIC approved *TrainDy* software for the calculation of longitudinal train dynamics (LTD) [17] and established the "special group" at that time consisting of Faiveley Transport (a Wabtec company), University of Rome Tor Vergata, DB Systemtechnik GmbH, SNCF Rolling Stock Engineering Centre (CIM), Trenitalia, and Knorr-Bremse SfS GmbH. The "special group" currently has 11 members representing RUs, industry, universities, and research institutes. *TrainDy* has been validated by comparing its numerical results against more than 30 experimental test trials made available by DB Systemtechnik GmbH, the SNCF Rolling Stock Engineering Centre (CIM), and Trenitalia. The main characteristic of *TrainDy* is its ability to solve both the pneumatic problem (venting of brake pipe and filling of brake cylinders [18]) and the mechanical problem (computation of the relative distance between adjacent wagons). Its calculation is currently limited to 1D dynamics; for an LTD simulation implementing multibody, see, for example, [19]. Despite this, the computational efficiency of *TrainDy* makes it particularly suitable for studying new trainset compositions by analyzing the risk of derailment (and train disruption) due to high LCF or longitudinal tensile forces (LTF) between the vehicles of the train.

It is worth noting that *TrainDy* has been used in several well-known European projects and many other industrial projects: in the European Union (EU) Framework Programme 7 with the MARATHON Project and in the EU Horizon 2020 Programme with the Shift2Rail Joint Undertaking Marathon2Operation project (www.marathon2operation.eu (accessed on 12 November 2022)).

This paper shows that *TrainDy* can also simulate trains equipped with a DAC, with varying degrees of complexity, to extract the maximum possible capacity from the current infrastructure safely. Section 2 illustrates the essential characteristics of the DAC; Section 3 provides a summary of how *TrainDy* handles electro-pneumatic brake and mechanical DAC (only with central coupler); Section 4 gives numerical results of trainsets with (a) radio only, (b) with mechanical DAC, (c) with mechanical and pneumatic DAC; Section 5 concludes the paper.

## 2. Digital Automatic Coupler (DAC)

The European DAC Delivery Program (EDDP) from Shift2Rail (S2R) is currently developing the Digital Automatic Coupler (DAC), intending to increase productivity in freight operations in a safe way. DAC is the cornerstone of several innovations mentioned in Europe's Rail Joint Undertaking (EU-Rail). Among these is the implementation of European

rail control-command and signalling (CCS), which is of central importance in running a safe, efficient, interoperable, robust, cost-efficient, and reliable rail service in Europe. DAC is part of the on-board CCS and is expected to significantly improve cross-border operations and multimodal customer services. DAC, among other characteristics, is an enabler for the European Train Control System (ETCS) moving block. It aims to transform the European rail freight system and establish fully digital freight train operations.

Depending on the DAC level (see Figure 1), DAC may reduce LCF generated in operated trains. In the figure, "AC 1" and "AC 2" refer to automatic couplers (AC) that are just "mechanical DAC" and which are referred to as "DAC2", in this paper. As a design requirement, any level of DAC will increase, and even double, the permissible LCF (PLCF) of the wagons to promote its implementation. "AC 1", "AC 2" and "DAC 3" are the same in terms of mechanical characteristics and PLCF. PLCF is currently assessed using the test methodology described in EN standard 15839 or the extrapolation rules cited in IRS 40421, which are valid for wagons equipped with classical (or side) buffers and drawgear, usually defined following EN 15551 and EN 15566 standards. Implementing DAC features will result in a significant leap forward in productivity by reducing the time required for shunting operations and extracting optimum capacity from the current infrastructure. DAC implementation applies train automation to its total capacity, e.g., automated yards, train preparation, operation and maintenance, and intermodal terminals.

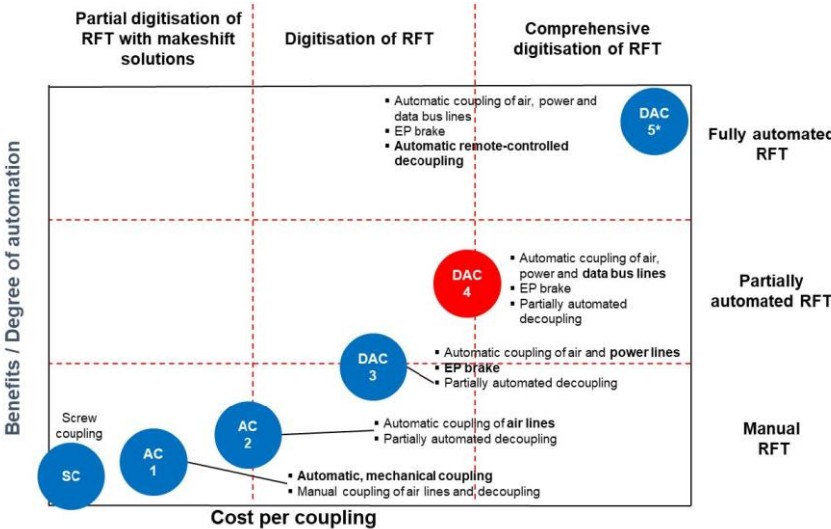

**Figure 1.** Different levels of train automation depending on DAC type.

As DAC is not yet in production, this paper presents an analysis of the first *TrainDy* simulation results, comparing the efficiency of the various technical means of optimizing freight transport by rail. The basis for this comparison is given by trains having length, hauled mass, and braking position, allowed by UIC IRS 40421.

## 3. Electro-Pneumatic Brake and Mechanical DAC

As quickly recalled in the paper introduction, *TrainDy* is a software that computes the LTD; therefore, each vehicle has just one degree of freedom. The vehicles exchange forces between them by the draw gear (when the wagons depart from each other) and the side buffers (when the wagons approach each other). These coupling components are modelled as non-linear friction elements with different loading and unloading curves. The mechanical DAC is modelled and simplified, as described at the end of this section. The calculation of air pressure in brake pipe requires the solution of a quasi-mono-dimensional fluid dynamic model, which considers the conservation of mass and energy and the balance of momentum ([18]). This model can study brake application and release. From the air pressure in the brake pipe, the air pressure in brake cylinders is computed via distributors'

transfer functions. The braking forces are computed from the air pressure in brake cylinders, in agreement with [20]. Adding the running resistance and the force from the traction units (whether they be traction or braking) is possible to compute the LTD: *TrainDy* is currently written in MATLAB and employs the ode solver ode15s.

One of the features of the DAC should be the ability to vent the brake pipe from several points along the train. This feature is somewhat like that of the electro-pneumatic brake on passenger trains. The current version of *TrainDy* does not natively support the EP brake, but it can be simulated utilizing a workaround: the wagons are substituted with fictitious traction units with the same wagon features in terms of mass, length, coupling and braked weight. Despite this, the *TrainDy* source code has been modified, assigning the EP brake feature (for braking only) to single wagons, replicating the TU's manoeuvre immediately in front of them. UIC Leaflet 541-5 [21] states for the EP brake feature that "*the pressure in the MAP [Main Air Pipe] shall be reduced from 5 bar to 3.5 bar over a period of 3.5 to 5 s*". To simulate the EP brake with *TrainDy*, the brake system is modelled as a lateral nozzle connected to the brake pipe of each wagon, with a diameter capable of venting the pressure locally, in compliance with UIC Leaflet 541-5. This diameter is a function of the time needed to reduce the pressure from 5 bar to 3.5 bar and the wagon length. The time to reduce the pressure from 5 bar to 3.5 bar is a function of the specific EP system. Figure 2 provides a diagram correlating the nozzle diameter to the vehicle length from 7 to 40 m (step 1 m) to reduce the pressure in the brake pipe from 5 bar to 3.5 bar in 4.25 s (this is the time used for this paper). This way, it is possible to associate a general vehicle with the corresponding nozzle diameter by interpolation.

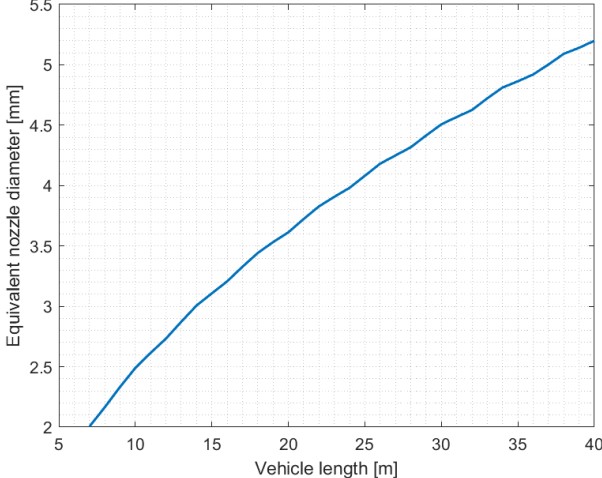

**Figure 2.** Nozzle diameter as a function of wagon length to reduce pressure from 5 bar to 3.5 bar in the brake pipe. The time for this reduction is 4.25 s.

A train consisting of seven identical wagons (or coaches), each contemporary discharging the brake pipe with the same nozzle, is used to draw the diagram in Figure 2. The given nozzle diameter can reduce the air pressure in the BP from 5 bar to 3.5 bar, at the middle of the 4th wagon, in 4.25 s. Considering a lower number of wagons in the train, e.g., three or five, and monitoring the pressure reduction in the 2nd and 3rd wagon, respectively, reduces the computational time needed to obtain a similar diagram, providing diameters slightly different (usually higher). Figure 3 shows this circumstance where the differences are at most of the order of 2%, and they are magnified just to show their behaviour. Increasing the number of wagons in the train consist provides a solution less dependent on un-avoidable numerical approximations. Moreover, since these numerical errors depend on the pressure wave reflections at the boundaries of the brake pipe, the errors reduce with the increase of wagon length.

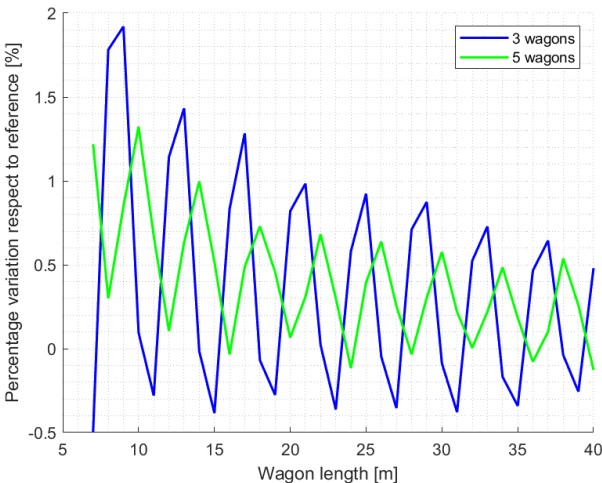

**Figure 3.** Variation of nozzle diameter with wagon length for different numbers of wagons in the trainset.

Polynomials can fit data reported in Figure 2; Table 1 reports the coefficients of these polynomials up to the 5th order, along with the commonly used parameters to evaluate a fitting: $R^2$ (R-squared) is a measure of how well the data fit the regression model (the goodness of fit); $R_{adj}^2$ (adjusted R-squared) shows whether adding additional predictor variables improves a regression model. Good regressions require the above coefficients towards 1. The table shows that a 4th order polynomial fit already provides a suitable fitting, since it has the highest $R_{adj}^2$.

**Table 1.** Fitting the diameter of the equivalent nozzle for EP brake, with a discharge time of 4.25 s.

| p5 | p4 | p3 | p2 | p1 | p0 | $R^2$ | $R_{adj}^2$ |
|---|---|---|---|---|---|---|---|
| | | | | 0.093575 | 1.6418 | 0.98598 | 0.98554 |
| | | | $-1.2335 \times 10^{-3}$ | 0.15155 | 1.0794 | 0.99913 | 0.99908 |
| | | $3.2178 \times 10^{-5}$ | $-0.003502$ | 0.19929 | 0.79261 | 0.99979 | 0.99977 |
| | $-1.3188 \times 10^{-6}$ | $1.5615 \times 10^{-4}$ | $-7.5465 \times 10^{-3}$ | 0.25246 | 0.56078 | 0.99987 | 0.99986 |
| $3.2922 \times 10^{-8}$ | $-5.1872 \times 10^{-6}$ | $3.2745 \times 10^{-4}$ | $-0.011078$ | 0.2859 | 0.44601 | 0.99988 | 0.99985 |

One final remark: even with the current *TrainDy* modification, the EP brake system discharges the brake pipe between the wagons. However, as of DAC level 3, the brake pipe discharge is applied between the connected wagons; this effect is expected to be minor, although it should be experimentally measured.

Within the scope of this paper, the mechanical DAC (AC2) is modelled using the force/stroke characteristics shown in Table 2. Since DAC is currently under development, the final design may be significantly different. It is important to remark that the same equivalent force/stroke characteristics are used for side buffers and draw gear coupling: the train's mechanical characteristics are the same whether DAC2 is used or not. This is a simplification introduced to better evaluate the effect of the increment of PLCF because of DAC. Consequently, see also Section 4.3, trains with EP brake and with DAC3 have the same mechanical features but different PLCF. The PLCF of wagons equipped with DAC has been increased conservatively by 50%. These figures should be confirmed by subsequent studies, which should evaluate how the PLCF changes as a function of the curvature radius, the wagon type, and the load, but they are considered realistic and conservative by the authors.

**Table 2.** Force/stroke characteristics of mechanical DAC.

| Stroke [mm] | 0 | 9 | 19 | 29 | 34 | 39 | 49 | 53 | 55 |
|---|---|---|---|---|---|---|---|---|---|
| Load [kN] | 50 | 70 | 130 | 260 | 370 | 490 | 900 | 1400 | 5000 |
| Unload [kN] | 35 | 49 | 91 | 182 | 259 | 343 | 630 | 980 | 3500 |

## 4. Numerical Results

This paper applies the "relative approach" described in the IRS 40421 [15]. The "relative approach" method compares a new train family with an accepted train family in terms of LCF by comparing the number of potential derailments of these two families. If this number is lower for the new train family, it can circulate safely. Trainsets respecting the IRS 40421 are examples of accepted train families, e.g., G-braked trains (all vehicles brake in position "G") having a mass of up to 4000 T and length up to 700 m (TU(s) excluded). An example of a new train family used in this paper is the one in which a radio-controlled traction unit works at the trainset end or a train having wagons equipped with a DAC.

The following train operations were used for the comparison: (i) emergency braking from full acceleration (TEB)—the train operation that is most challenging for LTD and not likely to occur; (ii) emergency braking (EB). The initial speed for all types of braking is 30 km/h, since the friction material of the braking shoes is cast iron (a conservative assumption), and the LTD is enhanced for braking applications from this speed.

For the sake of simplicity and since DAC studies are at an early stage, only one type of wagon is used for this paper to create the statistical trains; Table 3 shows its most relevant parameters. The wagon load can vary from zero (empty condition) to the maximum value (laden condition). Note that wagons equipped with a DAC will most likely have different characteristics, especially in terms of variation of braked weight with load: the choice of empty/loaded device is conservative since it is expected to result in higher LCF.

**Table 3.** Main wagon features.

| Name | Length [mm] | Tare [t] | Type | Braked Weight [t] | Shoes | Time to Reach 95/100% of Max Pressure in P [s] | Time to Reach 95/100% of Max Pressure in G [s] |
|---|---|---|---|---|---|---|---|
| Sdgkmss | 16.44 | 15.6 | Bogie | Loaded 53/Empty 23/Change 45 | 2 Bgu | 4/4.5 | 24/28 |

Since the time to reach the maximum pressure in brake cylinders (BC) can be different from the nominal behavior and because this time affects the LTD, this time has been randomly changed for each train-consist wagon to align the train simulations with the current wagons' behavior. Therefore, when the wagon brakes in passenger position (P) or goods position (G) [22], the time to reach 95% of maximum pressure in BC is uniformly distributed between 3 and 5 s or 22.5 and 25.5 s, respectively. The time to reach 100% of the pressure is obtained by multiplying the 95% time by 4.5/4 or by 28/24, respectively for trains operating in P and G.

This section is divided into four sub-sections. Section 4.1 reports the in-train forces of trains explicitly admitted by IRS 40421, providing a reference for DAC technology. To provide another reference for DAC technology and background for trains with mechanical DAC and train lengths up to 1500 m, Section 4.2 reports the probability of derailment for trains equipped with radio communication between the TUs compared to those of the worst trains admitted by the IRS 40421. Section 4.3 reports the probability of derailment when the wagons are equipped with a mechanical DAC, electro-pneumatic (EP) brake and DAC3 (i.e., a mechanical DAC + a brake similar to EP). Finally, considering trains with lengths up to 1500 m and made by coupling two trains, one with DAC2 and another without DAC, with TUs communicating by radio, this study addresses the better relative position of DAC2 trains.

### 4.1. Reference and Train Statistics

The reference train families for this paper are those explicitly allowed by UIC IRS 40421, i.e., trains with hauled mass in the range (1200–1600) T with Long Locomotive regime (LL) (the first five wagons and the leading TU are in freight position G, the remaining wagons are in passenger position P), and trains with hauled mass in the range (1600–2500) T and (2500–4000) T in G; all with train lengths in (640–740) m but without a DAC or EP brake. The first two train families are likely to have empty wagons, and this does not happen for the third train family because of the higher hauled mass. As the results of the following sections show, the higher minimum hauled mass of the last train family is beneficial in terms of safety.

Figure 4 part (a) shows a series of dots, each one representing the worst LCF (negative value) and LTF (positive value) for each train; as it is customary for these analyses, each train family is represented by 1000 trains, which is a number of trains that provides a satisfactory evaluation of mean and standard deviation of LCF (as required by [15]). Part (b) shows the ratios of LCF/PLCF and LTF/PLTF: in brackets, the labels show the percentages for potential derailment and train disruption, respectively; note that the permissible LTF is set to 550 kN, as this value prevents fatigue damage to the draw gear. Looking at (a) and (b), (2500–4000) T trains have LCF like (1200–1600) T trains in UIC IRS 40421 but have a lower risk of derailment because of higher wagon mass, as also confirmed by the histograms in (c) and (e). The high probability of potential derailment of the reference trains is the result of a conservative calculation: (i) the PLCF values in UIC IRS 40421 are conservative; (ii) the LTF/PLTF ratio assumes that the wagon subjected to the highest LCF is also running on a track section with a low radius of curvature; (iii) the likelihood of TEB is low (around $10^{-4}$).

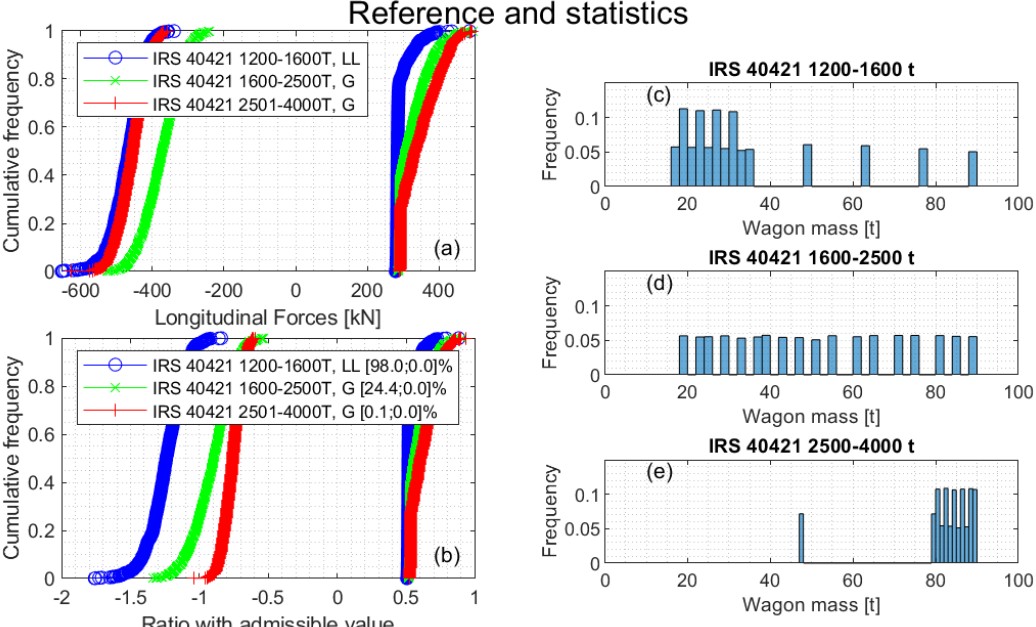

**Figure 4.** Reference trains without a DAC or EP brake. (**a**,**b**) report the cumulative frequencies of reference trains (see the legends) in terms of LF and Ratio with PLCF, respectively; (**c**–**e**) report the histograms of wagon mass for the three different train families (see the subfigures titles).

Note that the same trains are used and only the technology (or the brake position) is changed in the following sections: therefore, the effect of each change (e.g., radio or DAC) is enhanced. The next section shows the existence of other train families safer than those explicitly allowed to operate (taking as reference the (1200–1600) T in LL), as the aim of the "relative approach" methodology, envisaged by UIC IRS 40421, is to find new safe trainsets.

*4.2. Traction Units in Radio Communication*

Concerning radio communication between TUs, this paper reports the simulations of the LTD, when the train is formed by coupling two long trains with a total length of up to 1500 m, without or with a TU placed at the train end (the label "SW" is used in this case). Figure 5 illustrates the probability of potential derailment for UIC IRS 40421 (1200–1600) T trains in LL as blue and red vertical broken lines for TEB and EB, respectively. It also shows trains in different brake positions, where the TUs communicate via GSMR (a) or LTE (b). It is worth noting that the delay in radio communication between the TUs is considered randomly variable for each train. For trains equipped with GSMR radios, the reaction time of the guided TU relative to the leading TU is derived from a Gaussian distribution with a mean of 1.6 s and a standard deviation of 0.16/3 s. This time is the time required to discharge the brake hose at the point where the guided TU is located and is not just the communication time between the radios. For trains equipped with LTE radios, the corresponding Gaussian distribution has a mean of 0.9 s and a standard deviation of 0.045 s. The previous timings are in line with M2O Deliverable D3.3 [23] and they are conservative.

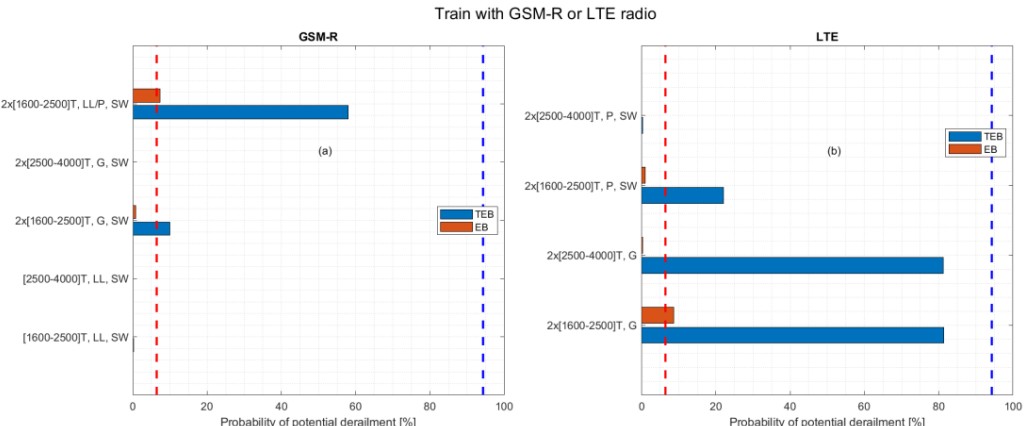

**Figure 5.** Trains with GSMR (**a**) or LTE (**b**) radio without DAC or EP brake in nominal mode (radio working correctly).

When a third TU is placed at the end of the trainset, there is a beneficial effect on the LCF because the BP is vented from the train end, and this stretches the train reducing the LCF. In this condition, to have a fair comparison with the same train family without the third TU at the end (e.g., 2 × (2501–4000) T), the acceleration phase for TEB operation is accomplished with 2/3 of the maximum power: thus, the train families employ the same power during the acceleration. In fact, the LTD is affected by the initial energy of the train before the EB application. Increasing the total power, having three TUs at their maximum power, the comparison between the train families in terms of in-train forces is also affected by the different initial potential energy (i.e., the couplers are more stretched or compressed). Therefore, the presence of the third TU is not the only difference in the comparison.

The train families 2 × (2500–4000) T show that it can safely increase the hauled mass up to 8000 t (with a train length up to 1500 m) using three TUs and with all vehicles in brake positions G and P, respectively, using GSM-R and LTE radio. Even if this figure shows several possibilities of safe trains with radio communication, they can be many more, e.g., reducing the overall length or coupling trains with different lengths and hauled masses. An example is given by 2 × (1600–2500) T, LL/P, SW, i.e., a train made by coupling two (1600–2500) T trains, the first in Long Locomotive (LL) and the second in passenger position (P) with a remote third TU; this is less safe than the reference UIC IRS 40421 (1200–1600) T in EB operation. A small reduction of the length and/or the mass of the second train makes this train family safer.

### 4.3. EP Brake and Mechanical DAC (DAC2)

This section demonstrates other possible ways of increasing the hauled mass by employing the EP brake (available also from DAC3 level) and mechanical DAC (DAC2). When a train is formed by coupling two trains and only the mechanical DAC (DAC2) is available, the two TUs must be radio-connected via GSM-R or LTE.

As in Figures 5 and 6 shows the probability of potential derailment of UIC IRS 40421 (1200–1600) T in LL with dashed vertical lines. Part (a) of Figure 6 shows that DAC2 enables the safe operation of single trains in the LL regime even if the hauled mass is more than 1600 t. As expected, implementing the EP brake solves the problem of train derailment for any hauled mass and length, as shown by the two train families "2 × (1601–2500) T, P EP" and "2 × (2501–4000) T, P EP" in part (b) of Figure 6.

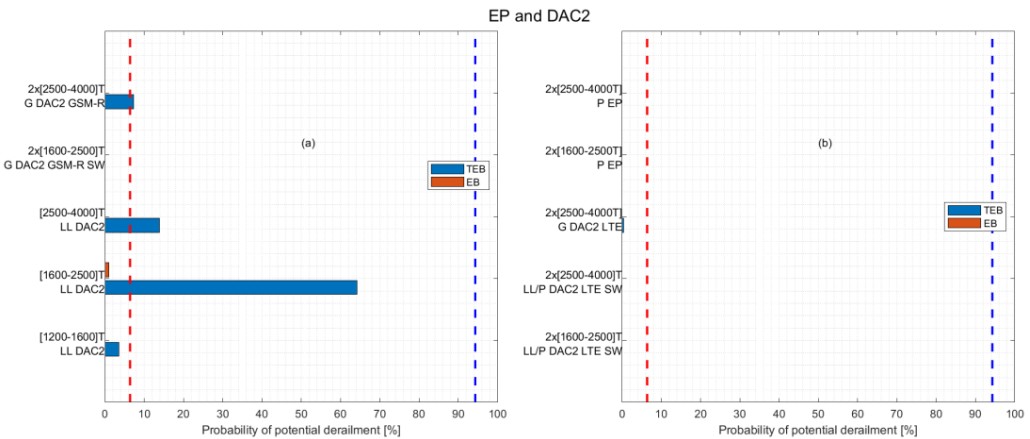

**Figure 6.** Trains with EP brake or DAC2.

A fundamental step forward is provided by DAC3, which is even more effective than EP, as illustrated in (a) and (b) of Figure 7. The part (a) of the figure shows that the LCF are the same, if EP or DAC3 are employed, since the couplers share the same mechanical features. The part (b) proves the beneficial effect of the higher PLCF provided by DAC3 in comparison with the side buffers used for simulation with just the EP brake: see the comparison between the green and blue curves or between the light blue and the red curves.

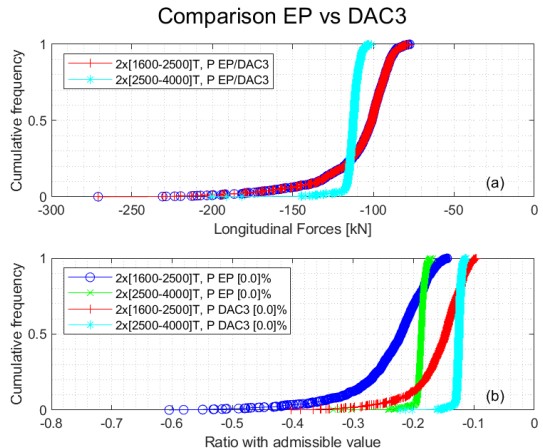

**Figure 7.** In-train forces for trains with EP or DAC3. (**a**,**b**) report the cumulative frequencies of EP and DAC3 trains in terms of LF and Ratio with PLCF, respectively.

### 4.4. Transition Trains

For the successful implementation of DAC, it is necessary to reduce as much as possible the transitory phase in which some trains have DAC and some others do not. Therefore, it

may be necessary to couple DAC and NoDAC trains and vice versa. Figure 8 clearly shows that the order of the two trains is irrelevant: in (a) the first sub–train is without DAC2 (NoDAC), which equips the second sub–train; in (b) it is vice versa. What is essential is the radio technology between the TUs (LTE is required to obtain an acceptable probability of derailment) and the braked positions (safe only in G). A relevant consequence of these results is that radio communication between the TUs is an essential feature to facilitate the migration, at least before the implementation of DAC3.

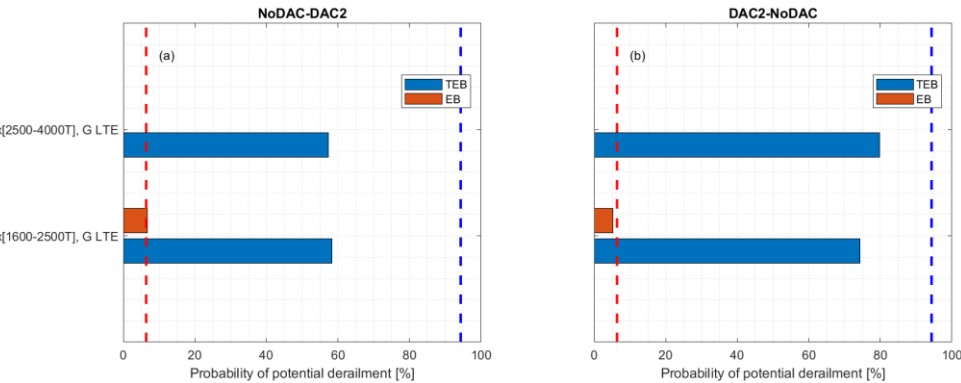

**Figure 8.** Coupling of DAC and NoDAC trains. In (**a**) the first sub–train is without DAC2, which equips the second sub–train; in (**b**) it is vice versa.

## 5. Conclusions

The paper focuses on several technologies that can be implemented very quickly, such as radio communication between traction units or within a relatively short period, such as the Digital Automatic Coupler (DAC), to increase freight efficiency and improve the railway's green footprint. Radio communication has recently been successfully tested by the Shift2Rail (S2R) FR8RAIL II and Marathon2Operation (M2O) projects. DAC is being studied and developed as part of several EU projects. This paper reports the in-train forces calculated by *TrainDy* software for different statistical train families. Based on the application of IRS 40421, these simulations give rise to the following conclusions: (a) radio communication between TUs is an effective way to increase the hauled mass both with GSM-R and LTE radio for single and coupled trains (especially if a third TU is placed at the end of the train); (b) mechanical DAC (DAC2) is an alternative way of increasing the hauled mass and using P trains instead of G trains; (c) with DAC2, the third TU is not required to couple two trains and arrive at a train length up to 1500 m; (d) DAC3 (i.e., including implementation of the electro-pneumatic brake) solves all problems with regard to in-train forces; and (e) employing DAC2 on just one of two coupled trains (in a transition phase, for example) requires the use of LTE radio and G trains.

**Author Contributions:** Conceptualization, R.T.; Methodology, G.R.; Software, L.C.; Formal analysis, A.O.; Data curation, T.D.; Writing—original draft, L.C.; Writing—review & editing, L.C.; Supervision, L.C. All authors have read and agreed to the published version of the manuscript.

**Funding:** This research received no external funding.

**Informed Consent Statement:** Not applicable.

**Data Availability Statement:** Not applicable.

**Conflicts of Interest:** The authors declare no conflict of interest.

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
