# Peer review of "The Digital Automatic Coupler (DAC): An Effective Way to Sustainably Increase the Efficiency of Freight Transport in Europe"

_sustainability, doi:10.3390/su142315671_

Round 1
Reviewer 1 Report
Dear Authors,
thank you for your work. I think that overall the paper is well organized and well written. I ask for some minor revisions
- 3rd paragraph of the introduction: it seems that an additional blank is present or something is missing in "the UIC IRS 40421 15. provides"...
- after a few lines: "train dynamics (LTD) [17. and", it seems a closing bracket is missing
- Section 2: it is written "Any level of 95 DAC will increase, and even double, the permissible LCF (PLCF) of the wagons." I think that a comment should be added at least to provide qualitative reasons for this expected behaviour.
- Section 3: I know TrainDY but, maybe, it would be nice to have its main features recalled: number of degrees of freedom, how buffers/draw gear or DAC are introduced, modelling of the pneumatic part and brake forces. It is just a matter of a few lines, to make the reader aware of the features of the model.
- what are the reasons for the oscillations in fig. 3? would it be possible to have also the data with 7 wagons plotted?
-honestly, I don't see the value added by table 1. I would simply say that numerical data can be suitably described with a polynomial of n-th order. Then you can list the parameters of the polynomial. The current caption of Table I does not sound immediate.
In Table 3 I think the length should be in meters.
in 4.2: "Concerning radio communication between TUs, this paper simulates"... the paper does not "simulate". Please, replace it with a proper subject.
- Consider the idea of splitting figure 5 into two parts. The quality of the figure is low and the text is hardly readable.
Author Response
The answers to the Reviewer are in the attached docx file

Reviewer 2 Report
This is an important and valuable research topic, but I have some suggestions for this paper. My detailed review is as follows:
1. The title does not point out the innovation of the paper.
2. In the introduction section, the annotations of the references cited in the paper are not uniform, such as reference 16.
3. It is written in the introduction that there are many ways to improve the efficiency of freight trains. It is recommended to introduce other related methods, compare the advantages and disadvantages, and then lead to the design direction of this paper to make the paper more convincing.
4. This paper does not specifically analyze the data reflected in the figure, as shown in Figure 4.
5. The "relative method" cited in the paper lacks specificity.
6. This paper selects only one type of carriage to create a statistical train, and selecting several types for data comparison can make the paper more rigorous.
7. A general introduction to the subsections of Chapter 4 is missing prior to Section 4.1.
8. There are some formatting errors in the references section.
Author Response
The answers to the Reviewer are in tha attached docx file

Reviewer 3 Report
A well-written paper that focuses on several new technologies in order to increase freight efficiency and improve the railway's green footprint. I suggest the paper be published in the form it is.
The Figure 5 is not clearly readable
Author Response

(The authors gave the same response as above.)

Round 2
Reviewer 2 Report
This is an important and valuable research topic, but I have some suggestions for this paper. My detailed comments are as follows:
1. The format of references is disordered, and the order is marked incorrectly. Please check and modify one by one.
2. The order of the pictures is repeated, please check and modify the full text.
Author Response
Dear Reviewer, we have answered in the attached file
